# Viewing Natural vs. Urban Images and Emotional Facial Expressions: An Exploratory Study

**DOI:** 10.3390/ijerph18147651

**Published:** 2021-07-19

**Authors:** Marek Franěk, Jan Petružálek

**Affiliations:** Faculty of Informatics and Management, University of Hradec Králové, Rokitanského 62, 500 03 Hradec Králové, Czech Republic; jan.petruzalek@uhk.cz

**Keywords:** face reading technique, facial emotional expressions, natural environment, positive emotions

## Abstract

There is a large body of evidence that exposure to simulated natural scenes has positive effects on emotions and reduces stress. Some studies have used self-reported assessments, and others have used physiological measures or combined self-reports with physiological measures; however, analysis of facial emotional expression has rarely been assessed. In the present study, participant facial expressions were analyzed while viewing forest trees with foliage, forest trees without foliage, and urban images by iMotions’ AFFDEX software designed for the recognition of facial emotions. It was assumed that natural images would evoke a higher magnitude of positive emotions in facial expressions and a lower magnitude of negative emotions than urban images. However, the results showed only very low magnitudes of facial emotional responses, and differences between natural and urban images were not significant. While the stimuli used in the present study represented an ordinary deciduous forest and urban streets, differences between the effects of mundane and attractive natural scenes and urban images are discussed. It is suggested that more attractive images could result in more pronounced emotional facial expressions. The findings of the present study have methodological relevance for future research. Moreover, not all urban dwellers have the possibility to spend time in nature; therefore, knowing more about the effects of some forms of simulated natural scenes surrogate nature also has some practical relevance.

## 1. Introduction

People react to the natural environment mostly with positive emotions, and viewing the natural environment also has a positive function in mental restoration [1,2,3]. Moreover, there is a large body of evidence that exposure to simulated natural scenes also has similar positive effects [4]. Viewing simulated natural scenes may help people to improve their actual emotional state and their mental restoration in situations in which they have no opportunity to visit a real natural environment. Thus, it is useful to deeply analyze immediate emotional reactions to various types of natural scenes with diverse research methods. In the present study, we explored emotional facial expressions in viewing natural and urban images and employed automated facial expression analysis by machine vision software. These techniques have developed and improved considerably in the last three decades and may overcome the drawbacks and time consumption associated with the facial action coding system and the technical difficulties of facial electromyography. Recently, automated computer-based technologies have demonstrated sufficient reliability (e.g., [5,6]), and their accuracy may surpass that of human raters in many cases [7].

### 1.1. Positive Effects of Viewing Surrogate Nature

A growing body of research has documented the therapeutic and health-improving effects of contact with the natural environment. Many studies have shown that being in a natural environment benefits health, increases positive emotion, reduces stress, and has direct and positive impacts on well-being and mental health (for review, [1,2,3]).

However, not all urban dwellers have the possibility to spend time in nature; moreover, the natural environment may be difficult to access from large metropolitan areas. Therefore, environmental psychologists have explored whether some forms of simulated nature can have at least partially similar positive effects. The research findings have suggested that exposure to simulated natural scenes (e.g., viewing photographs, slides, videos, and virtual computer-generated nature scenes (for review, see [4])) may also have similar positive effects. For instance, it was documented that viewing natural images can improve mood and perceived restoration [8,9,10,11,12]. Viewing natural videos can also improve mood and perceived restoration and reduce stress (e.g., [13,14,15,16,17,18,19,20]). A similar effect was revealed with the exploration of nature scenes in virtual reality (e.g., [21,22,23,24,25]). Some of these studies used self-reported assessments (e.g., [8,11,14,16,20,26]), while others used physiological measures or combined self-reports with physiological measures (e.g., [13,15,17,18,22,27,28]). A detailed overview of methods surveying environmental perception was provided by Browning et al. [21]. An analysis of facial emotional expression has only rarely been used (e.g., [29,30,31,32,33]).

### 1.2. Ekman’s Six Basic Emotions

The link between emotions and facial expressions is based on theoretical grounds and has empirical support. Several decades ago, Ekman [34] defined the six most basic emotions that should be common in all cultures. They are anger, disgust, fear, happiness, sadness, and surprise. They can be easily recognized in facial expressions. Although they are also other theoretical frameworks that describe emotions in a dimensional space (e.g., [35]), Ekman’s concept of six emotions has preferentially been used in analyses of facial expressions of emotions.

### 1.3. Measurement of Emotional Facial Expressions

Currently, three methods are used in research studies to measure facial expressions of emotion: the facial action coding system, facial electromyography, and automatic computer facial expression analysis. The first method, the facial action coding system, is based on a subjective identification of six basic emotions in video-recorded faces [36]. Specially trained human coders evaluate specific emotional expressions called “action units” that account for the expression of six basic emotions. The action unit is the smallest visible functional facial movement that human observers can recognize. Although this is a method that provides sufficient validity in facial emotion description, its disadvantage is the considerable time required for data processing. Another research technique, facial electromyography, is based on monitoring activations of facial muscles during changes in emotional responses. It requires applying electrodes on the skin surface. It enables the identification of the specific facial muscle patterns used to display, for instance, joy, appetite, and disgust (e.g., [37]). This technique enables the detection of subtle facial muscle activities, but its disadvantage is technical complexity. Moreover, having electrodes attached to the face is far from a natural condition.

### 1.4. Validation of Software for Automated Facial Expression Analysis

Recently, there have been three commercial major software tools for automated facial expression analysis: Noldus’ FaceReader (Noldus Information Technology, Wageningen, The Netherlands) [38], iMotions’ FACET module (iMotions, Copenhagen, Denmark) [39], and iMotions’ AFFDEX module (iMotions, Copenhagen, Denmark) [39]. There is a debate regarding the reliability of these software programs in emotion recognition compared to facial electromyography or the facial action coding system. In a comparison with EMG results, Beringer et al. [40] validated the software FACET for happy and angry expressions. Recently, Kulke et al. [41] compared AFFDEX emotion recognition software with facial electromyography measurements for the ability to identify happy, angry, and neutral faces. However, there might be specific situations where human observers are better than automated face analysis. For instance, Del Líbano et al. [42] investigated how prototypical happy faces can be discriminated from blended expressions with a smile but nonhappy eyes and found that human observers using facial action units were better than those using FACET software for automated analysis. They concluded that the software FACET can be a valid tool for categorizing prototypical expressions, but it is not reliable enough for the discrimination of blended expressions.

### 1.5. Facial Expressions While Viewing Natural Environment

There are only a few studies that analyzed facial expressions when viewing the natural environment. These studies mostly employed facial electromyography. Electromyographic responses were mostly measured using the facial muscles of the forehead because these muscles can reflect mental and emotional stress better than other muscles. An increase in facial electromyography amplitude is a reflection of an increased level of muscle tension and, conversely, a decrease in amplitude reflects decreased tension.

Cacioppo et al. [29] presented slides that were mildly to moderately evocative of a positive and negative effect (mountain cliff, bruised torso, ocean beach, and polluted roadway) for 5 s to participants. They found that facial electromyographic activity over the brow, eye, and cheek muscle regions differentiated the pleasantness and intensity of affective reactions to the visual stimuli. In the study by Chang et al. [30], participants viewed images of an office with window views of nature or the urban environment for 15 s. The electrodes were placed above the eyebrows. The amplitude of electromyography, whose growth indicated an increasing degree of muscle tension, was inspected. In addition, changes in EEG, blood volume pulse, and state anxiety were recorded. The results indicated that the participants were less anxious, while watching a view of nature or indoor plants in contrast to offices without window views or offices without plants. However, the electromyographic results were inconsistent with the other measures. While there were lower amplitudes with the city-window office, the highest amplitude, curiously, was with the nature-window office. In a subsequent study, Chang et al. [31] presented natural images to the participants with various levels of restorativeness, each for 10 s. Electromyographic responses were measured using the facial muscles of the forehead, and EEG and blood volume pulse were assessed. The results revealed a large degree of congruency between the psychological measures of restorativeness and the three physiological responses. In summary, these few studies suggested that viewing natural images may elicit changes in facial expressions.

To date, automated facial expression recognition has not been used for the analysis of facial movements in viewing urban images. However, Wei et al. [32] explored facial emotional expressions in a real outdoor environment during a walk. Participants were asked to repeatedly take selfies while walking on urban streets or in a forest park that reflected their natural facial expressions and real-time emotions. The photographs were analyzed using FireFACE software. It was shown that the forest experience evoked higher happy scores but lower neutral scores than the urban environment.

### 1.6. The Goals

To date, our knowledge about emotional reactions after viewing images with natural environments registered via facial expressions is rather limited. To our knowledge, this technique has not been used in the context of environmental psychology and environmental preference research. Our goal was to explore these direct facial expressions while viewing a diverse range of images by using automated facial expression analysis, specifically iMotions’ AFFDEX software. In the present study, facial expressions while viewing natural images, namely, forest trees with foliage, forest trees without foliage, and urban images were investigated. Based on previous findings, it was hypothesized that natural images would evoke a higher magnitude of positive emotions in facial expressions and a lower magnitude of negative emotions than urban images. Furthermore, we explored whether people react in a different way to forest trees with foliage and forest trees without foliage

## 2. Materials and Methods

### 2.1. Participants

Sixty-six undergraduates participated in the experiment. The sample comprised young adults between the ages of 18 and 25 (mean = 20.97, SD = 1.11; 42 females). The participants were enrolled in the first, second, or third year of various psychology courses. They were students in informatics, financial management, and tourism at the University of Hradec Králové. The University of Hradec Králové is a small regional university, and the students come mostly from nearby, the northeastern regions of the Czech Republic—Hradec Králové and Pardubice. In this area, there are mostly lowlands or temperate highlands, mostly with deciduous forests. The participants lived in towns and villages, where the natural environment is easily accessible. Thus, the stimuli presented in the experiment (see below) included the type of landscape known to the participants. Similarly, the types of city buildings were known to the participants.

### 2.2. Ethical Approval

Ethical approval for the present study was obtained from the Committee for Research Ethics at the University of Hradec Králové (No. 4/2018). Participants signed an informed consent form in which they declared that they voluntarily participated in the experiment and that they were informed about the experimental procedure. They agreed that recordings of their facial behavior would be registered and used for scientific purposes only. They were allowed to withdraw from the experiment at any time.

### 2.3. Stimulus Material

Images used in the experiment were taken by one of the authors (Figure 1). They included images of forests and urban scenes. The images were transformed into a 1920 × 1080 pixel resolution using Adobe Photoshop CS 6 software. All images had their brightness levels and contrast balanced using the “Auto Levels”, “Auto Contrast”, and “Auto Colors” options in Adobe Photoshop. The photographs were not further digitally modified. Twenty-four images were presented in one experimental session. Eight natural images of deciduous forests with foliage were taken mainly in forests along the city of Prague. An additional eight natural images of deciduous forests without foliage were taken in the same areas as the previous set of photographs. Eight images were photographs of urban streets in Prague in the Czech Republic.

### 2.4. Apparatus

The experiment was controlled by a PC computer with a 1920 × 1200 pixel resolution screen and a diagonal of 61 cm with a Logitech Webcam C920 camera (Logitech, Newark, CA, USA) that was situated on the top of the screen. The camera and presentation of stimuli, as well as the data processing, were controlled by the software iMotion 8.0. The facial expression analysis was conducted by iMotions Facial Expression Analysis Module AFFDEX (iMotions, Copenhagen, Denmark). The web camera recorded facial videos while the participants viewed the stimuli, and then, videos were imported into the iMotions software for facial expression analysis postprocessing. AFFDEX enables the measurement of seven emotional categories: joy, anger, surprise, fear, contempt, sadness, and disgust. All emotional indicators were scored by the software on a scale from 0 to 100, indicating the probability of having detected the emotion. A magnitude of 0 indicated that the emotion was absent; in turn, a magnitude of 100 indicated a 100% probability of having detected the emotion.

### 2.5. Procedure

The participants were tested individually in a laboratory. The research was conducted in December 2019 within working days from December 10 to December 18 from 9:00 to 16:00. The participants selected the date and time of the experimental session according to their free time. After arrival to the laboratory, the participant signed the informed consent form. Then, he/she was informed about the experiment and read the instructions. The instructions were as follows: “You will take part in a study, in which you will successively examine a series of images presented on the computer screen. View an image with composure. Do not try to remember its content or its details. Your face will be recorded. Each image will be displayed for 15 s”. The participants sat approximately 70 cm from the display monitor. The images were presented in a random order. Every trial started with a fixation cross situated in the center of the screen on a gray background. The participants had to fixate on the fixation cross for 2 s before the image appeared. Each image was displayed for 15 s. There was a comfortable temperature in the laboratory, about 23 degrees Celsius.

## 3. Results

First, the raw data were exported from AFFDEX. Approximately, 240 measurements were obtained for one image, and approximately, 1900 measurements were obtained for one participant within one image category (urban images, forest images with vegetation, forest images without vegetation, see Appendix A). Next, the mean scores were calculated for each participant and the images in each category (Table 1). The results showed that the level of identified facial emotions was very low, under 1%, and differences between the scores for specific emotions under these conditions were also small. One-way repeated measures analyses of variance (ANOVA) were conducted to test the effect of the experimental condition (urban images, forest image with vegetation, forest images without vegetation) on the level of facial expression of specific emotions (Table 2). The analyses showed that the effect of the experimental conditions was nonsignificant for facial expressions of all emotions. It was only for facial expressions of the emotion fear, compared to facial expressions of other emotions, where more pronounced differences were found between urban images and both sets of forest images in the expected direction; however, the *p*-value was only 0.121.

## 4. Discussion

By using automated facial expression analysis, the present study explored whether a short viewing of urban or natural environments would elicit changes in facial expressions of emotions that might reflect changes in actual emotional state. Although we predicted differences between facial expressions while viewing urban and natural images, we did not find any significant differences in our study, which is in contrast with a large body of previous research (for review, see [21]), which documented diverse reactions to virtual urban and natural scenes by using introspection or different physiological methods.

In the study by Wei et al. [32], participants walked along a forest or an urban street for five hours and were asked to take selfies every 30 min by posing with their natural facial expressions and real-time emotion. Photographs of their faces were analyzed and processed by facial expression analysis software to obtain scores for happy, sad, and neutral expressions. It was found that the forest walk evoked higher happy and lower neutral expressions than the walk in an urban environment. Clearly, people who spend a long time in a pleasant natural environment might express positive emotion on their faces. Thus, the first explanation of our failure may be that the 15-second viewing of an image was too short to elicit observable facial expressions of an emotional reaction. However, Cacioppo et al. [29] observed that even five seconds of presentation of slides with outdoor environments resulted in changes in facial expressions; however, they used a different measure, namely, facial electromyographic activity.

The second possible explanation may be that the visual stimuli used in the present study were not sufficiently distinct to elicit intense emotional reactions accompanied by visible emotional facial expressions. As examples of urban images, we used photographs of ordinary urban apartment houses from the first half of the 20th century. Similarly, natural images represented photographs of ordinary deciduous forests located around the capital city taken under “normal” atmospheric conditions. Moreover, they were not further digitally modified to make them more attractive. Thus, the visual stimuli used in this experiment represented common environments where participants were living and, thus, may not have had the capacity to elicit a feeling of pronounced emotional responses. In the present study, we did not employ attractive natural images that have mostly been used in environmental psychology research [20], such as high mountains, rocks, lakes, sea, etc. For instance, Cacioppo et al. [29], who reported changes in facial electromyographic activity after 5-second slide presentations of natural stimuli, used stimuli that were rated as mildly to moderately pleasant (e.g., mountain cliff) or mildly relaxing (e.g., ocean beach). Clearly, a mountain cliff or an ocean beach are more distinct environments than central European lowland forests. This explanation is consistent with the findings of the Joye and Bolderdijk study [43], where participants watched pictures of awesome and mundane nature. They found that watching awesome natural scenes compared to mundane nature scenes and a neutral condition had pronounced emotional effects. Clearly, future research should compare the effects of mundane vs. more attractive natural scenes on emotional facial expressions. It is worth commenting on possible effects of participants’ experiences and cultural background on perception and estimation of aesthetical values of natural environments (e.g., see scenic beauty estimation method [44]). These individual variables may even result in a different estimation of the scenic beauty of an identical natural environment. Moreover, although verbal evaluations of environments also have their cognitive component that may be influenced by common beliefs (e.g., nature is beautiful, an urban street is ugly), facial emotional expressions are more spontaneous and reflect actual emotions. Thus, these individual variables may play a more substantial role in our research than in investigations based only on the verbal estimation of the environment. A specific environment may elicit positive emotion because people may have positive experiences and memories with that environment, or the environment is surprising because it is in strong contrast with their everyday environment, and this may elicit a desire to visit such an attractive environment, etc. Thus, the further limitation of the present study is that these individual experiences and backgrounds were not explored.

Furthermore, we may also speculate that the experimental situation, when participants know that they are part of the experiment and quickly observe diverse visual stimuli, may also differ from real-life situations, when they are using some form of virtual nature for relaxation. On the other hand, a large body of research has observed various emotional reactions to the natural environment in the laboratory (for review, see [4]). Perhaps an appropriate instruction that stresses the necessity to concentrate on visual stimuli and to imagine that they are inside the environment for a relaxed walk may strengthen emotional reactions.

As mentioned above, to date, there is a lack of data from studies that used the same research methodology and computer software. Our results can only be compared with the data obtained in the most recent study that was conducted in a different field. Otamendi and Sutil Martín [45] explored facial expressions in perceiving video advertisements processed by the same AFFDEX software that was used in our study. In their study, the participants viewed advertisement spots lasting 91 s that consisted of 31 scenes. The spots showed the accompanying role that a mother plays throughout the life of a child, from birth to adulthood. Similarly, they also reported small values for specific emotions, the highest for joy with a mean = 4.82, and smaller for the other emotions with means between 0.42 and 1.12 (AFFDEX scores emotions on a scale from 0 to 100). Only in the target group for the advertisement (mature aged women) did they find higher emotional reactions (mean for joy = 14.17). Thus, their investigation obtained similar small average values for emotional facial expressions in nontarget groups, as we found in our study.

To conclude, although our findings did not confirm differences in emotional reactions to natural and urban scenes, we suppose that there are other variables that may influence these findings. A low emotional salience of the pictures was already mentioned. Moreover, it seems that a random and short presentation of different visual stimuli, which is typical for experiments in the area of visual perception, is not ideal for the investigation of emotional reactions to visual stimuli, even where they consist of “mundane” environments and are not emotionally salient enough. It seems that by using more attractive visual environments, it could be possible to find significant differences in facial emotional expressions. Moreover, instructions in the experiment to be more immersed and engaged in the presented visual environment may affect the results. It is also worth commenting on possible individual variables. Thus, the present study has methodological relevance for future research. Moreover, knowing more about the effects of viewing simulated natural scenes on emotional reactions also has practical relevance.

## 5. Conclusions

The present study represents one of the first attempts to use automated facial expression analysis by machine vision software within the context of environmental psychology and research on preferred environments. The results showed that a mundane environment with low emotional salience did not elicit significant facial emotional expressions. This finding may help future research that could provide deeper insights into the positive Sileffect of viewing certain forms of simulated natural scenes. Not all urban dwellers have the possibility to spend time in nature, or the natural environment may be difficult to access. Therefore, it is useful to explore whether some forms of simulated nature can have at least partially similar positive effects.

## Figures and Tables

**Figure 1 ijerph-18-07651-f001:**
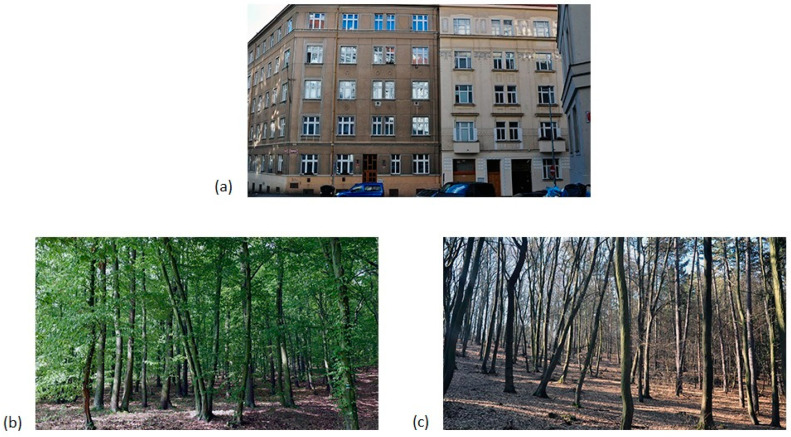
Examples of stimuli used in the experiment: (**a**) urban image, (**b**) forests with foliage, and (**c**) forests without foliage.

**Table 1 ijerph-18-07651-t001:** Mean scores for individual emotional categories with exposure to urban images, forest images with foliage, and forest images without foliage (the scale ranged from 0 to 100).

Emotion	Urban Scenes		Forest with Foliage		Forest without Foliage	
	Mean	SD	Mean	SD	Mean	SD
Anger	0.210	0.819	0.106	0.346	0.160	0.571
Contempt	0.249	0.204	0.257	0.324	0.326	0.444
Disgust	0.483	0.152	0.522	0.413	0.474	0.160
Fear	0.219	0.743	0.126	0.442	0.126	0.525
Joy	0.143	0.789	0.098	0.675	0.143	0.768
Sadness	0.288	1.076	0.264	0.905	0.265	0.781
Surprise	0.468	1.760	0.370	1.116	0.330	0.914

**Table 2 ijerph-18-07651-t002:** Results from one-way repeated measures ANOVAs for individual emotional categories.

Emotion	*df*	F	*p*
Anger	2, 128	1.112	0.332
Contempt	2, 128	1.558	0.214
Disgust	2, 128	0.728	0.485
Fear	2, 128	2.148	0.121
Joy	2, 128	0.050	0.608
Sadness	2, 128	0.123	0.884
Surprise	2, 128	1.640	0.200

## Data Availability

The datasets supporting this article have been uploaded as part of the Appendix A.

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
