# Peer review of "Viewing Natural vs. Urban Images and Emotional Facial Expressions: An Exploratory Study"

_ijerph, 2021, doi:10.3390/ijerph18147651_

Round 1

Reviewer 1 Report

The theme proposed in the research is interesting and relevant.
However, the authors used simple and perhaps unsuitable elements to achieve the desired goals.
The suggestion is for a more elaborate statistical analysis to be carried out, and for images and software reading values to be related.
The way the result was presented makes interpretation difficult for the reader.

Author Response

Dear reviewer, thanks for time devoted to reading  our manuscript and for your comments. However, we do not understand very well to your criticism and suggestion, maybe there may have been some misunderstanding. Therefore, in our answers we tried to explain briefly our position.

Point1: the authors used simple and perhaps unsuitable elements to achieve the desired goals.

Response 1: The goal of our study was to investigate emotional facial expressions while viewing urban and natural images. Therefore, we analyzed emotional facial expressions by iMotions’ AFFDEX software designed for the recognition of facial emotions. Thus, we believe that we used appropriate elements to achieve the desired goals.

Point2: The suggestion is for a more elaborate statistical analysis to be carried out, and for images and software reading values to be related.

Response 2: Both raw data (included in the supplementary material) and descriptive statistics show that facial emotional expression are very small. To explain the method of measurement - the software performs repeated measurements of facial muscles movements in period about 60 msec and in each period express the probability of having detected an emotion on a scale from 0 to 100. We found probabilities of having detected the emotion bellow 1.  From these low probabilities of emotion detections, no more sophisticated statistical method would get better results.

Point3: The way the result was presented makes interpretation difficult for the reader.

Response 3: The results were presented in a standard form in tables and in standard form of description the statistical analyses used. Moreover, the raw data are available.

Reviewer 2 Report

The paper presents interesting assumptions for using facial expression measuring. However, the research design is auspicious, needs more explanation to become clear and understandable. 

The abstract is clear and interesting. My suggestion is only to consider using the term "surrogate" because you examined a real landscape, not a surrogate of it. Maybe I understand it the wrong way, so an additional definition is needed. 

All these questions or answer can make light on the results. I like the discussion, but the last sentence (lines 303-304) seems to be out of the topic.

The introduction is quite long but well structured and easy to read. Lacking review of methods of surveying environment perception would be nice. The worse is the description of material and methods. I have few questions about it. The firsts are about the group of respondents: was it one group examined at the same time? Or each faculty separately? What environment? Time of the day? Were they rest or tired after few hours of classes? Where were the students from, and in what kind of landscape they have grown? The second set of questions is about the method. It is quite similar to Scenic Beaty Estimation (SBE)(Daniel, Boster 1976). Can You write more about the limitations of the method? It is your only one step in research design, so it has to be reliable.

Conclusions are too laconic. 

Author Response

Dear reviewer, thank you for your comments.

Point1: The abstract is clear and interesting. My suggestion is only to consider using the term "surrogate" because you examined a real landscape, not a surrogate of it.

Response 1: We replaced the term to “simulated natural scenes“.

Point2: Lacking review of methods of surveying environment perception would be nice.

Because we did not want to prolong introduction, we added reference to the study, where these methods were described in details, lines 61-62 “Detailed overview of methods surveying environmental perception was provided by Browning et. al [21].“  

Point 3: The firsts are about the group of respondents: was it one group examined at the same time? Or each faculty separately? What environment? Time of the day? Were they rest or tired after few hours of classes? Where were the students from, and in what kind of landscape they have grown?

Response 3: Maybe that we did not clearly specify that the experiment was conducted in the laboratory and that participants were tested individually. It was now specified on lines 203-205 “The participants were tested individually in a laboratory. The research was conducted in December 2019 within working days from December 10 to December 18 from 9:00 to 16:00. The participants selected the date and time of the experimental session according to their choice in their free time.”

Point 4: Where were the students from, and in what kind of landscape they have grown?

Response 4. It was specified on lines 163-169: “The University of Hradec Králové is a small regional university and the students come mostly from nearby, the Northeastern regions of the Czech Republic – Hradec Králové and Pardubice. In this area there are mostly lowlands or temperate highlands, mostly with deciduous forests. The participants lived in town and villages, where natural environment is easily accessible. Thus, the stimuli presented in the experiment (see below) included the type of landscape known to the participants. Similarly, the types of city buildings were known to the participants.”

Point 5: The second set of questions is about the method. It is quite similar to Scenic Beauty Estimation (SBE)(Daniel, Boster 1976). Can You write more about the limitations of the method?

Response5: We added the following discussion, see lines 284-297. Scenic Beauty Estimation was quoted, and possible limitations was explained: “It is worth commenting on possible effects of participants’ experiences and cultural background on perception and estimation of aesthetical values of natural environments [e.g., see Scenic Beauty Estimation Method, 44]. These individual variables may even result in a different estimation of the scenic beauty of an identical natural environment. Moreover, although verbal evaluations of environments have also their cognitive component that may be influenced by common beliefs (e.g., nature is beautiful, an urban street is ugly), facial emotional expressions are more spontaneous and reflect actual emotions. Thus, these individual variables may play in our research more substantial role than in investigations based only on the verbal estimation of the environment. A specific environment may elicit positive emotion because people may have positive experiences and memories with that environment, or the environment is surprising because is in strong contrast with their everyday environment, may elicit a desire to visit such attractive environment, etc. Thus, the further limitation of the present study is that these individual experiences and backgrounds were not explored“.

Point 6: I like the discussion, but the last sentence (lines 303-304) seems to be out of the topic.

Response 6: The sentence was removed.

Point 7: Conclusions are too laconic. 

Response 7: The conclusions were expanded.

Round 2

Reviewer 1 Report

The manuscript is very well written e it is an important contribution.

Author Response

Dear reviewer, thank you very much for your effort and positive evaluation of our work.